# Leveraging Evidence-Guided LLMs to Enhance Trustworthy Depression Diagnosis

Yining Yuan[1†], J. Ben Tamo[1†], Micky C. Nnamdi[1], Yifei Wang[1], May D. Wang[1]

[1]*Georgia Institute of Technology*, Atlanta, USA

{yyuan394, jtamo3, mnnamdi3, ywang4343, maywang}@gatech.edu

*Abstract*—**Large language models (LLMs) show promise in automating clinical diagnosis, yet their non-transparent decision-making and limited alignment with diagnostic standards hinder trust and clinical adoption. We address this challenge by proposing a two-stage diagnostic framework that enhances transparency, trustworthiness, and reliability. First, we introduce evidence-guided diagnostic reasoning (EGDR), which guides LLMs in generating structured diagnostic hypotheses by interleaving evidence extraction and logical reasoning, grounded in DSM-5 criteria. Second, we propose a Diagnosis Confidence Scoring (DCS) module that evaluates the factual accuracy and logical consistency of generated diagnoses through two interpretable metrics: Knowledge Attribution Score (KAS) and Logic Consistency Score (LCS). Evaluated on the D4 dataset with pseudo-labels, EGDR outperforms Direct in-context prompting and Chain-of-Thought (CoT) across five LLMs. For instance, on OpenBioLLM, EGDR improves accuracy from 0.31 (Direct) to 0.76 and DCS from 0.50 to 0.67. On MedLlama, DCS rises from 0.58 (CoT) to 0.77. EGDR yields up to +45% accuracy and +36% DCS gains over baselines, offering a clinically grounded, interpretable foundation for trustworthy AI-assisted diagnosis.**

*Index Terms*—**Large Language Models, Diagnostic Reasoning, Mental Health AI, DSM-5, Knowledge Graphs, Explainable AI, Evidence-Guided Reasoning.**

## I. Introduction

Depression is a widespread and debilitating mental health disorder [1], affecting approximately 280 million people globally, according to the World Health Organization (WHO), and contributing significantly to the global burden of disease [2]. WHO further estimates that depression and anxiety together cost the global economy over $1 trillion each year in lost productivity, driven by factors such as absenteeism, reduced work performance, and unemployment [2]. These figures underscore the urgent need for improved strategies in early detection and diagnosis of depression, as timely intervention can significantly reduce morbidity and save lives [3].

Recent advances in large language models (LLMs) offer promising tools for analyzing textual data and identifying subtle linguistic markers of psychological distress [4], [5]. However, despite their fluency and general reasoning capabilities, most LLM-based diagnostic tools suffer from a lack of transparency [6]. They often function as black-box classifiers, directly mapping input text to diagnostic labels without providing interpretable justification [7], [8]. This limitation critically undermines clinical reliability, as medical decision-making requires interpretable, evidence-based justification. A recent scoping review in digital pathology emphasizes that one of the major hurdles to safely deploying large language models (LLMs) in medical settings is their limited ability to provide contextualized and interpretable outputs tailored to specific clinical scenarios [9]. In addition, broader literature points out that merely offering diagnostic predictions is often inadequate for clinical use, as the opaque reasoning behind LLM-generated results can weaken clinician trust, which underscore the need for explainable reasoning in diagnostic tools [7].

We propose a two-stage diagnostic pipeline designed to improve the trustworthiness, transparency, and clinical alignment of LLM-generated diagnoses. In the first stage, the Evidence-Guided Diagnostic Reasoning (EGDR) framework guides LLMs to interleave evidence extraction and logical inference, producing structured diagnostic hypotheses. Using multi-turn patient-clinician dialogues and the DSM-5 [10] as sources of medical knowledge, EGDR aligns reasoning with formal diagnostic criteria to enhance interpretability and clinical grounding. We evaluate EGDR using pseudo-labels from a strong baseline model and further validate its performance on MDDial [11], a secondary dataset with gold-standard annotations.

In the second stage, our DCS framework evaluates diagnosis reasoning from first stage via micro-level claim attribution and macro-level reasoning consistency. While prior claim verification methods usually treat claims in isolation [12], our approach jointly assesses the full reasoning chain. DCS combines two components: the Knowledge Attribution Score (KAS), inspired by ClaimVer [13], which checks alignment with DSM-5 triplets, and the Logic Consistency Score (LCS), which verifies rule-based criteria like symptom thresholds and exclusions. Together, they offer a comprehensive, interpretable measure of diagnostic validity, supported by alignment with pseudo-label accuracy and case-level analysis.

## II. Related Works

The evolution of clinical diagnostic reasoning has increasingly embraced LLMs, shifting away from rule-based and traditional machine learning approaches towards flexible, generative systems. Prompt engineering, retrieval-augmented generation (RAG), fine-tuning, and domain-specific pre-training have emerged as prominent methods for interpretable diagnoses across a range of medical conditions.

Prompt-based methods use crafted inputs or few-shot ex-

---

[†] The first two authors contributed equally to this work.

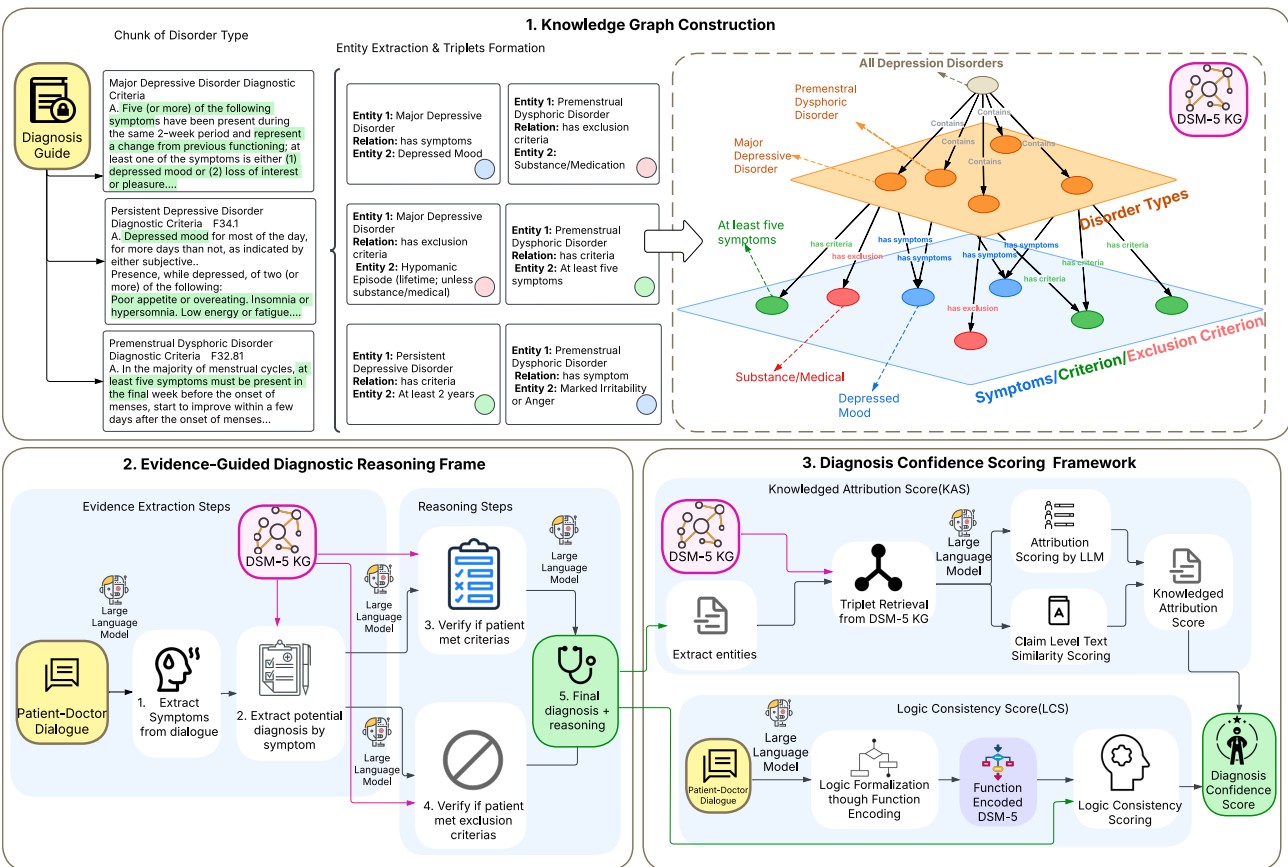

Fig. 1: Overview of the two-stage Evidence-Guided Diagnostic Reasoning (EGDR) and Diagnosis Confidence Scoring (DCS) framework. A DSM-5-based knowledge graph is constructed from diagnostic manuals using medical triplets. Stage 1 (EGDR) processes multi-turn patient dialogues to extract symptoms, retrieve relevant criteria, and generate diagnosis with reasoning. Stage 2 computes two evaluation scores: Knowledge Attribution Score (KAS) for semantic alignment with DSM-5 triplets, and Logic Consistency Score (LCS) for rule-based diagnostic validity. These are aggregated into a final Diagnosis Confidence Score (DCS) ranging from 0 to 1.

emplars to guide reasoning without additional training. Wu et al. [14] introduced Diagnostic-Reasoning Chain-of-Thought (CoT), significantly improving diagnostic accuracy by emulating physician reasoning. Kwon et al. [15] further advanced reasoning-awareness by prompting explicit clinical rationales before diagnoses. Nachane et al. ClinicR [16] applied iterative CoT strategies for open-ended medical questions, achieving superior accuracy.

Fine-tuning LLMs for depression diagnosis specifically has shown notable success. Zhang et al. [17] develop a dialogue-based system combining ChatGLM-6B with a multi-step diagnostic pipeline and knowledge-aware prompting, trained on the D4 dataset to output symptom summaries and risk assessments. ChatGLM-based models focus on severity scoring rather than subtype classification. Li et al. [18] propose DepLLM, a modular mixture of specialized experts approach to handle complex depressive symptom clusters, albeit language-specific and lacking direct DSM-5 interpretability. In parallel, causal reasoning can enhance the trustworthiness of clinical AI. For instance, Tamo et al. [19] show that incorporating causal inference into predictive frameworks improves individualized treatment-effect estimation in high-stakes surgical settings [20].

To improve factual correctness, Wang & Shu [21] introduced

FOLK, a First-Order Logic approach that grounds model outputs through symbolic claim decomposition. Liu et al. [22] proposed retrieval-augmented frameworks that integrate structured medical knowledge to reduce hallucinations, while Dmonte et al. [12] surveyed LLM claim-verification methods spanning multi-hop reasoning, semantic entailment, and hybrid symbolic-neural verification. Yet most methods still assess claims in isolation, overlooking global logical consistency. A recent review [23] found that fine-tuned or ensemble transformers, including BERT, RoBERTa, DistilBERT, and GPT-3/4 with methods like prefix-tuning and domain-specific pretraining, consistently outperformed traditional models in detecting, classifying, and managing depression from social media, EHR, and clinical data.

Despite these gains, current systems remain limited. Prompting strategies (CoT, DoT) lack mechanisms for global diagnostic coherence, fine-tuned models focus narrowly, and retrieval or symbolic methods seldom integrate formal diagnostic logic or offer interpretable reasoning. Reliance on surface metrics (accuracy, F1) further constrains evaluation. These gaps highlight the need for hybrid frameworks that couple LLMs with clinical knowledge and symbolic reasoning for more reliable, interpretable depression diagnosis.

## III. METHODOLOGY

Our method employs a two-stage diagnostic framework as depicted in Figure 1:

1) **Evidence-Guided Diagnostic Reasoning (EGDR)** extracts and structures diagnostic reasoning from patient–doctor dialogues, grounding it in DSM-5 criteria.
2) **Diagnosis confidence scoring** verifies these claims against a DSM-5–based knowledge graph.

Our DSM-5 knowledge graph ($KG_{\text{DSM-5}}$) builds on the general principles of medical knowledge graphs described by Wu et al. [24], with extensions to capture DSM-5-specific entities and relations. We extract entity–relation–entity triplets from DSM-5 text (e.g., "Major Depressive Disorder (MDD) $\rightarrow$ has symptom $\rightarrow$ depressed mood"). A root node links all disorders, with directed edges such as "has_criterion", "has_exclusion", and "has_specifier" to capture relationships among disorders, symptoms, and criteria, enabling efficient clinical reasoning.

### A. Evidence-Guided Diagnostic Reasoning (EGDR)

Given a multi-turn dialogue between a patient and psychologist/clinician, $D = \{(u_1, r_1), \ldots, (u_n, r_n)\}$, where $u_i$ and $r_i$ are the patient and doctor utterances at turn $i$.

We prompt an LLM $f_\theta$ with dialogue context $D$ using EGDR, Figure 2, which guides the model to produce structured diagnostic output $R_{LLM}$.

Formally, the model output is structured as:

$$R_{LLM} = f_\theta(\text{prompt}(D, KG_{\text{DSM-5}})) \qquad (1)$$

This process yields a semi-structured final diagnosis and diagnostic reasoning $C$, that consist of analysis on symptoms, criterion, exclusion criterion, and final diagnosis.

### B. Diagnosis Confidence Score Calculation

To evaluate the generated diagnosis hypotheses, we propose Diagnosis Confidence Scoring (DCS), which decomposes into two components, knowledge attribution score (KAS) and logic consistency score (LCS).

#### a) Knowledge Attribution Score (KAS)

KAS measures the factual alignment of individual diagnostic claims with the DSM-5 knowledge graph $KG_{\text{DSM-5}}$, combining symbolic matching and neural semantic similarity to assess whether each claim is clinically valid. This design builds on prior work in natural language claim verification and medical fact attribution [13], adapting it to the diagnostic setting where claims must be grounded in medical guidelines.

In $KG_{\text{DSM-5}}$, each diagnosis represents a central node connected to nodes representing symptoms, exclusion rules, specifiers, and modifiers. Edges encode semantic relationships such as "has symptom," "has exclusion," or "has criteria," allowing graph-based retrieval of relevant knowledge for each claim.

The KAS pipeline consist of:

- **Entity Extraction and Triplet Retrieval**. We extract psychiatric entities from the reasoning passage $R$ using medical named entity recognition (NER), then retrieve relevant triplets $T(R) \subset KG_{\text{DSM-5}}$ via WoolNet [25], a graph-walk strategy that incrementally traverses clinically relevant subgraphs using a hybrid of symbolic reasoning and neural relevance scoring, enhancing both interpretability and reasoning accuracy in LLM-based diagnosis.

- **Claim Decomposition and Verification**. Diagnosis reasoning $R$ is decomposed into atomic, verifiable claims $\{c_1, c_2, \ldots, c_n\}$ using an LLM. Each claim $c_i$ is evaluated for factual alignment by comparing it against the retrieved DSM-5 triplets $T(R) \subset KG_{\text{DSM-5}}$ from last step.

- **Triplet-Based Attribution Classification**. Each claim $c_i$ is classified based on its symbolic alignment with retrieved triplets $T(R) \subset KG_{\text{DSM-5}}$, and assigned a symbolic score $cs(c_i)$ as follows:

$$cs(c_i) = \begin{cases} 2 & \text{Attributable} \\ 1 & \text{Extrapolatory} \\ -1 & \text{Contradictory} \\ 0 & \text{No Attribution} \end{cases}$$

- **Triplets Match Score (TMS)**. Each claim $c_i$ is also assigned a Triplets Match Score :

$$TMS_i = \alpha \cdot \text{sim}(c_i, T(R)) + (1 - \alpha) \cdot EPR_i \qquad (2)$$

where $\text{sim}(c_i, T(R))$ is the cosine similarity between the claim and retrieved triplets using a BERT-based encoder, $EPR_i$ the entity-level precision/recall between $c_i$ and matched DSM entities, and $\alpha \in [0, 1]$ is a weighting hyperparameter, we set it to 0.5, through analyzing TMS and EPR distribution and experimenting a set of configuration and finding a suitable trade-off balance between TMS and EPR.

- **Claim Score Composition**. A weighted trust score for each claim by combining symbolic classification and semantic similarity:

$$w_i = cs(y_i) \cdot TMS_i \qquad (3)$$

- **KAS Aggregation.** We apply a sigmoid function that adjusts sensitivity based on the aggregate evidence weight. This allows the KAS to effectively distinguish flawed reasoning from valid justifications.

$$KAS = \sigma\left(\sum_{i=1}^{n} w_i\right), \quad \text{where } \sigma(x) = \frac{1}{1 + e^{-x}} \qquad (4)$$

#### b) Logic Consistency Score (LCS)

While KAS evaluates the factual accuracy of individual claims, LCS assesses whether the overall reasoning aligns with the full diagnostic logic of the DSM-5. Each DSM-5 disorder criterion is encoded as an executable function that explicitly captures the required symptom count, presence of core symptoms, and any exclusionary conditions. The LLM is prompted to execute a step-wise application of these diagnostic rules over its reasoning trace. The LCS score is then assigned

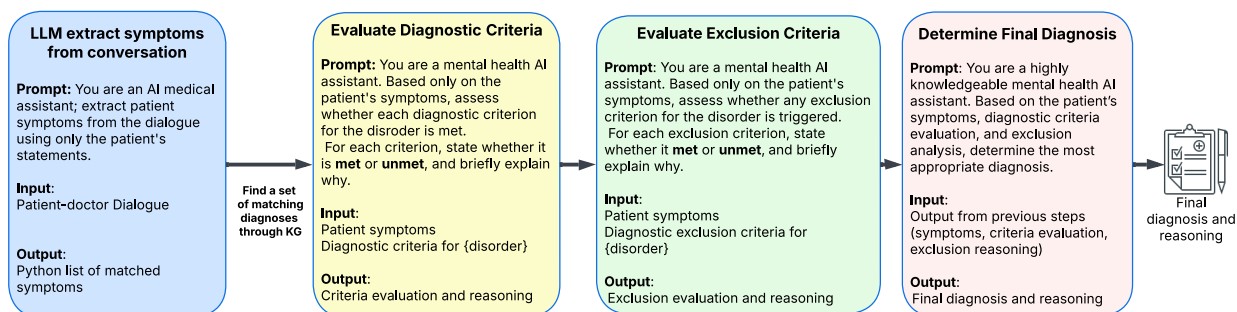

Fig. 2: EGDR Framework Overview: (1) An AI assistant extracts patient symptoms from doctor-patient dialogue; (2) Matches symptoms to top-3 candidate disorders via the DSM-5 knowledge graph; (3) Evaluates whether diagnostic criteria are met; (4) Checks exclusion criteria; (5) Determines the final diagnosis and provides reasoning.

based on how closely the reasoning conforms to the structured diagnostic criteria.

$$LCS = \begin{cases} 0, & \text{Contradicts DSM-5 diagnostic logic} \\ 1, & \text{Partially correct logic, incorrect conclusion} \\ 2, & \text{Partially correct logic, correct conclusion} \\ 3, & \text{Fully correct logic and conclusion} \end{cases}$$

The final DCS combines KAS and LCS, using a normalized LCS score scaled to $[0, 1]$:

$$DCS = \lambda \cdot KAS + (1 - \lambda) \cdot (\frac{LCS}{3}), \quad \text{where } \lambda \in [0, 1] \quad (5)$$

This unified score provides a normalized, interpretable measure of diagnostic reliability from 0 (invalid) to 1 (fully valid).

## IV. EXPERIMENTS

We evaluate our framework across a range of LLMs to assess diagnostic quality and reasoning fidelity in depression diagnosis and suicide risk assessment. To isolate the effect of our clinical evidence-guided reasoning pipeline, we include both domain-specific and general-purpose models. Experiments were conducted using a mix of local and cloud resources: local evaluations ran on dual NVIDIA A100 GPUs, while API-based models (e.g., Claude, GPT-4o-mini, Gemini) were accessed as available.

We begin by benchmarking all models on the Dialogue Dataset for Depression-Diagnosis (D4) [26], using a four-class depression and suicide risk scoring task to establish baseline performance and identify the top model for generating diagnostic pseudo-labels. This model is then used to create silver-standard labels for the D4 dataset by applying DSM-5 criteria to ground truth symptom annotations, which serve as evaluation targets for all models. In addition to the D4 dataset, to further demonstrate the effectiveness of our EGDR reasoning pipeline, we conducted experiments on the MDDial dataset [11], which includes a different set of diagnoses and gold-standard labels.

To assess the impact of our reasoning pipeline, we compare it against two prompting baselines: (1) direct instruction prompting, which elicits structured diagnostic outputs, and (2) chain-of-thought prompting, which encourages stepwise reasoning. All approaches include DSM-5 criteria, but our pipeline uniquely integrates them in a structured, diagnostic-oriented manner,

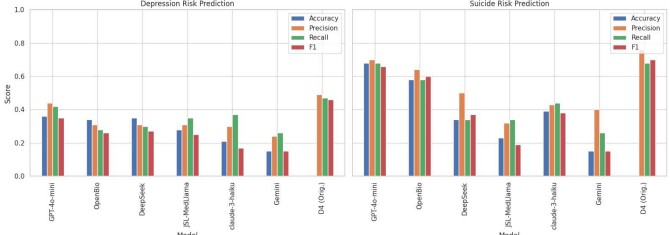

Fig. 3: Depression and suicidal risk prediction result. GPT-4o-mini performance approaches D4 baseline

systematically grounding reasoning in the formal framework. Baselines use the same information but with less structured guidance.

Finally, we apply our DCS framework to evaluate factual accuracy and logical validity, analyzing alignment with pseudo-labels, per-case reasoning quality, and the contribution of individual scoring components via ablation. These results demonstrate the robustness and interpretability of the DCS method.

### A. Dataset

We use the D4 dataset [26], which contains 1,339 annotated clinician–patient dialogues focused on mental health evaluation, including expert-labeled symptom mentions, depression severity, and suicide risk. The dataset is slightly imbalanced, with a risk score of 0 dominating both tasks. These annotations enable supervised evaluation of both diagnostic performance and reasoning quality.

Based on benchmarking results (Figure 3), GPT-4o-mini was selected as the pseudo-label generator for downstream tasks due to its strong alignment with expert risk scores and superior performance among evaluated models.

Alongside evaluating the D4 dataset, we use the MDDial dataset [11], which comprises 235 patient dialogue instances in its test set. This dataset offers authoritative labels across a variety of diagnoses, such as Esophagitis, Enteritis, Asthma, Coronary Heart Disease, Pneumonia, Rhinitis, Thyroiditis, Traumatic Brain Injury, Dermatitis, External Otitis, Conjunctivitis, and Mastitis. We assessed a selection of models on this dataset, illustrating the versatility and robustness of our proposed framework in handling both general and disease-specific clinical dialogue tasks.

TABLE I: Performance by Model and Prompting Method evaluated on GPT-4o-mini pseudo labels. DCS score represents the average Diagnosis Confidence Score.

| Model | Prompting | Acc | Prec | Rec | F1 | DCS |
|---|---|---|---|---|---|---|
| MedLlama [27] | EGDR | 0.68 | 0.79 | 0.68 | 0.73 | 0.77 |
| | Direct | 0.65 | 0.71 | 0.83 | 0.77 | 0.72 |
| | CoT | 0.69 | 0.85 | 0.69 | 0.70 | 0.58 |
| Claude [28] | EGDR | 0.77 | 0.83 | 0.77 | 0.80 | 0.73 |
| | Direct | 0.75 | 0.82 | 0.75 | 0.78 | 0.69 |
| | CoT | 0.70 | 0.91 | 0.70 | 0.78 | 0.59 |
| DeepSeek [29] | EGDR | 0.29 | 0.86 | 0.29 | 0.30 | 0.56 |
| | Direct | 0.21 | 0.86 | 0.21 | 0.32 | 0.44 |
| | CoT | 0.11 | 0.76 | 0.12 | 0.20 | 0.31 |
| OpenBioLLM [30] | EGDR | 0.76 | 0.81 | 0.76 | 0.75 | 0.67 |
| | Direct | 0.31 | 0.77 | 0.31 | 0.44 | 0.50 |
| | CoT | 0.46 | 0.84 | 0.46 | 0.59 | 0.50 |
| Gemini [31] | EGDR | 0.88 | 0.85 | 0.86 | 0.86 | 0.77 |
| | Direct | 0.79 | 0.81 | 0.79 | 0.80 | 0.70 |
| | CoT | 0.76 | 0.85 | 0.76 | 0.80 | 0.74 |

TABLE II: Performance of different models and prompting methods on MDDial dataset.

| Model | Prompting | Accuracy | Precision | Recall | F1 Score |
|---|---|---|---|---|---|
| Claude | EGDR | 0.69 | 0.77 | 0.69 | 0.71 |
| | Direct | 0.57 | 0.45 | 0.46 | 0.46 |
| | CoT | 0.57 | 0.55 | 0.56 | 0.56 |
| Gemini | EGDR | 0.75 | 0.74 | 0.74 | 0.75 |
| | Direct | 0.71 | 0.71 | 0.72 | 0.72 |
| | CoT | 0.69 | 0.36 | 0.31 | 0.33 |
| DeepSeek | EGDR | 0.48 | 0.15 | 0.16 | 0.16 |
| | Direct | 0.21 | 0.15 | 0.17 | 0.17 |
| | CoT | 0.41 | 0.15 | 0.18 | 0.18 |

## B. Diagnosis, Evaluation, and Confidence Score

As shown in Table I, the evidence-guided reasoning prompting consistently outperforms both Direct and Chain-of-Thought (CoT) prompting across most evaluation metrics, including accuracy, recall, and F1 score across all models. The most substantial improvement is observed in OpenBioLLM, where EGDR achieves a 0.45 gain in accuracy over Direct prompting (0.76 vs. 0.31). Notably, even strong general-purpose models like Claude and Gemini benefit from EGDR, with Gemini reaching the highest accuracy overall (0.88) under EGDR. These results demonstrate that integrating structured clinical evidence and diagnostic reasoning steps significantly enhances classification performance, regardless of model domain specialization.

On the MDDial dataset (Table II), EGDR similarly outperforms both Direct and CoT prompting across most models. Gemini shows the highest overall performance with EGDR (Accuracy: 0.75, F1: 0.75), while Claude also benefits significantly (F1: 0.71 with EGDR vs. 0.46 with Direct). Even with lower-performing models like DeepSeek, EGDR consistently improves results over baseline prompting strategies. These findings reinforce EGDR's robustness and adaptability in handling complex, multi-turn clinical dialogues.

Our analysis further demonstrates that DCS functions effectively as a diagnostic confidence metric. As illustrated in Figure 4, DCS scores for correct cases are consistently higher across prompting strategies, while incorrect cases receive markedly lower scores, indicating strong diagnostic confidence.

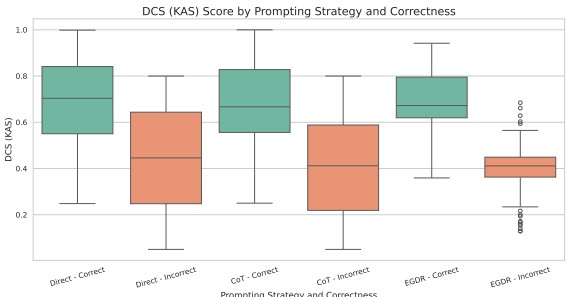

Fig. 4: Diagnosis Confidence Scores (DCS) by correctness under different prompting strategies.

Notably, EGDR yields a more concentrated distribution of DCS values for correct and partially correct outputs, with scores tightly clustered around higher values. This suggests that EGDR improves reasoning structure and calibrates confidence more reliably. In contrast, Direct and CoT prompting tend to produce more vague or unsupported reasoning, resulting in wider and more dispersed DCS distributions with greater overlap between correct and incorrect cases.

## C. Fairness Analysis Across Age and Gender

TABLE III: Average accuracy across all models and prompting methods by demographic group.

| | ≤17 | 18–25 | 26–35 | 36–45 | 46–60 | 60+ | Men | Women |
|---|---|---|---|---|---|---|---|---|
| Avg Acc | 0.635 | 0.616 | 0.558 | 0.496 | 0.495 | 0.364 | 0.528 | 0.578 |

To assess the fairness of large language model (LLM) reasoning across demographic groups, we conducted a subgroup performance analysis by age and gender. The dataset itself is notably imbalanced. In terms of age, the 18–25 group dominates with 685 samples, followed by 26–35 (307), 36–45 (138), 46–60 (105), ≤17 (100), and 60+ (4). Gender distribution is similarly skewed, with 982 female and 357 male samples.

Because the LLM was not trained or fine-tuned on this dataset, dataset imbalances do not affect the model parameters directly. Instead, the subgroup disparities observed in performance reflect two main factors: (1) evaluation reliability—performance estimates are more stable in larger subgroups (e.g., 18–25), while very small groups such as 60+ yield volatile results where a few errors can drastically change accuracy; and (2) potential pretraining and reasoning biases within the LLM itself, which may influence responses differently across demographic attributes.

Overall, models show consistently higher accuracy for the majority subgroups (18–25, 26–35, and female participants), while smaller or minority groups exhibit lower or more variable performance, shown in Table III. This highlights the importance of cautious interpretation of fairness metrics in zero-shot LLM settings, where disparities may arise from a combination of intrinsic model biases and uneven group representation in the evaluation data.

## D. Impact of α, and λ on DCS

We performed an ablation study, Table IV, to explore how two key parameters, $\alpha$ and $\lambda$, shape the behavior and interpretability of the Diagnostic Consistency Score (DCS). Rather than focus-

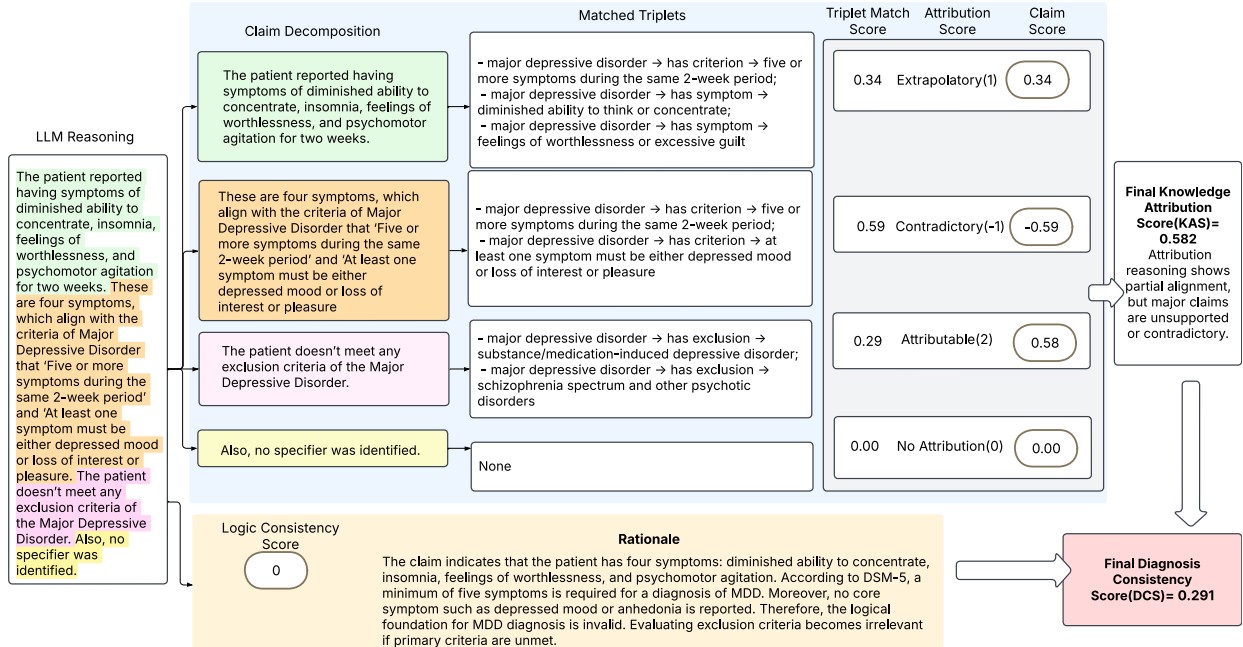

Fig. 5: DCS evaluation of diagnostic reasoning for an invalid MDD case. The top section shows a Knowledge Attribution Score (KAS) breakdown, revealing mostly weak alignment with DSM-5 diagnostic knowledge. The bottom section presents the logic consistency evaluation, which scores 0.0 due to missing core symptoms and insufficient symptom count. Together, these produce a low Diagnostic Consistency Score (**DCS = 0.291**).

TABLE IV: Ablation study showing how variations in $\alpha$ and $\lambda$ affect the distribution of DCS. $\alpha$ controls the trade-off between semantic similarity and entity-level precision/recall in the Triplet Match Score (TMS), while $\lambda$ balances KAS and LCS in the final DCS. The table reports summary statistics across reasoning outputs, highlighting the impact of each parameter on scoring behavior and stability.

| Param | Value | Mean | Std Dev | Min | 25% | Median | 75% |
|---|---|---|---|---|---|---|---|
| $\alpha$ | 0.00 | 0.3517 | 0.2790 | 0.0085 | 0.1471 | 0.2171 | 0.6422 |
| | 0.25 | 0.7454 | 0.2274 | 0.0083 | 0.7510 | 0.8153 | 0.8768 |
| | 0.50 | 0.8462 | 0.1160 | 0.0258 | 0.8219 | 0.8534 | 0.9197 |
| | 0.75 | 0.8653 | 0.1129 | 0.0094 | 0.8459 | 0.8614 | 0.9331 |
| | 1.00 | 0.8811 | 0.0781 | 0.2583 | 0.8559 | 0.8623 | 0.9362 |
| $\lambda$ | 0.00 | 0.5700 | 0.2726 | 0.0000 | 0.4500 | 0.4500 | 0.7500 |
| | 0.25 | 0.6621 | 0.2069 | 0.0086 | 0.5743 | 0.5859 | 0.8074 |
| | 0.50 | 0.7541 | 0.1501 | 0.0172 | 0.6981 | 0.7215 | 0.8648 |
| | 0.75 | 0.8462 | 0.1160 | 0.0258 | 0.8219 | 0.8534 | 0.9197 |
| | 1.00 | 0.9382 | 0.1250 | 0.0152 | 0.9409 | 0.9600 | 0.9820 |

ing solely on numerical performance, our analysis highlights how these parameters modulate core trade-offs in semantic attribution and reasoning fidelity.

### a) Alpha $\alpha$ – Balancing Semantic Similarity and Entity-Level Precision/Recall in TMS.

The $\alpha$ parameter controls the balance in the Triplet Match Score (TMS) between semantic similarity (SS) and entity-level precision/recall (EPR). Higher $\alpha$ emphasizes SS, which captures embedding-level closeness between claims and retrieved knowledge triplets. However, strong SS alone does not ensure that a claim is attributable (A) or extrapolating (E); it may still reflect contradiction (C) or no attribution (N). In contrast, EPR enforces stricter, symbolic alignment at the level of clinical entities, but may miss paraphrased or inferred matches.

Thus, $\alpha$ controls how sensitive TMS is to surface-level fluency versus grounded clinical precision. High $\alpha$ values lead to higher TMS and trickle into higher KAS and DCS scores, but this can result in overconfident scoring for plausible-sounding but incorrect claims. Lower $\alpha$ brings EPR to the forefront, reducing false positives but potentially penalizing semantically valid variation. We choose $\alpha = 0.5$ to maintain a balance: enough semantic flexibility to reward paraphrased reasoning, but enough entity grounding to detect extrapolation or misattribution.

### b) Lambda $\lambda$ – Balancing KAS and LCS.

The $\lambda$ parameter controls the weighting between KAS (claim-level attribution to retrieved knowledge) and LCS (global diagnostic logic consistency) in the final DCS score. High $\lambda$ values prioritize KAS, rewarding models that justify individual claims well, even when their overall diagnostic reasoning is flawed. For instance, at $\lambda = 1$, the mean DCS reaches 0.9382, but this can reflect locally plausible yet globally inconsistent outputs. Conversely, low $\lambda$ values favor LCS, emphasizing logical coherence. At $\lambda = 0$, the mean DCS drops to 0.57, showing that LCS alone is brittle. It penalizes valid reasoning that deviates from expected rule structures and underweights attribution.

To balance attribution fidelity and logical soundness, we set $\lambda = 0.75$, which achieves a strong mean DCS of 0.8462 with low variance. This choice helps prevent models from being rewarded for individually plausible claims that, when combined, fail to justify the overall diagnosis.

### E. Case Study: Low Diagnosis Confidence Scoring

The case study presented in Figure 5 demonstrates the effectiveness and interpretability of the DCS. In the low confidence case, the DCS score of 0.291 reflects both limited knowledge grounding (KAS = 0.582) and a failed symbolic logic check (Logic Score = 0.0). Although some symptoms are recognized, the claims do not meet the MDD threshold, with only four

symptoms and no core symptom identified. This breaks a core diagnostic requirement outlined in the DSM-5, making the reasoning incomplete. This inconsistency is accurately penalized in both the semantic and logic evaluations. By aligning both structured diagnostic knowledge and rule-based logic, DCS ensures that the diagnosis is grounded not only in claim correctness but also in rigorous diagnostic logic.

## V. Conclusion

This work proposes a unified two-step framework for LLM-based diagnostic reasoning: (1) the Evidence-Guided Diagnostic Reasoning (EGDR) pipeline, which enhances output quality through clinically grounded prompting, and (2) the Diagnosis Confidence Score (DCS), a dual-layered metric evaluating both factual alignment and diagnostic logic. Experiments on the D4 and MDDial datasets show that EGDR consistently improves diagnostic performance across diverse models. The best-performing model, Gemini-2.5-flash, achieved an 8% accuracy improvement on D4 and a 4% accuracy improvement on MDDial with EGDR. Our DCS evaluation also demonstrates strong alignment with silver-standard labels on D4, and case analyses confirm its ability to generate clinically coherent and well-grounded reasoning. Together, EGDR and DCS advance the transparency, reliability, and clinical fidelity of AI-assisted diagnosis.

### Acknowledgement

This research was supported in part through research cyberinfrastructure resources and services, including the AI Makerspace of the College of Engineering, provided by the Partnership for an Advanced Computing Environment (PACE) at the Georgia Institute of Technology, Atlanta, Georgia, USA. We also gratefully acknowledge funding and fellowships that contributed to this work, including a Wallace H. Coulter Distinguished Faculty Fellowship, a Petit Institute Faculty Fellowship, and research funding from Amazon and Microsoft Research awarded to Professor May D. Wang.

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
