# OpenReview forum: "Leveraging Evidence-Guided LLMs to Enhance Trustworthy Depression Diagnosis"
_IEEE.org/EMBS/BHI/2025/Conference — BHI 2025_

### Official Review · Reviewer_qDZP · 2025-07-01
**Leveraging Evidence-Guided LLMs to Enhance Trustworthy Depression Diagnosis**

**Confidence:** 4
**Clarity Of Writing:** great
**Clinical Significance:** good
**Methodological Novelty:** good
**Overall Rating:** 6
**Final Rating:** 7

**Experiments And Results:**

great

**Questions For The Authors:**

Thank you for submitting your work to BHI 2025, I have the following comments and questions:
1. DSM-5 seems to be a large manual that consists of hundred thousands of tokens (words). In Section IV, you mentioned that all approaches include DSM-5 criteria. Could you extend on how is the DSM-5 criteria being injected into baseline methods? Is the whole DSM-5 criteria being used by the baseline methods?
2. For the claim decomposition and verification, how to determine the number of n? If the n is large, won't that lead to several incomplete claims (resulting in a symbolic score of 1 instead of 2?). In addition, how is the symbolic score (-1, 0, ..) being determined?
3. Judging by the F1 score presented in Table I, it's hard to decide whether EDGR performs better than other baselines. For example, Direct performs the best (or equallly best with CoT) with MedLlama, Deepseek and Gemini; EDGR only performs the best on Claude and OpenBioLLM. Also, the F1 calculation for EDGR on Gemini seems odd, with Acc=0.86, Prec=0.85 and Rec=0.856, the F1 score is only 0.65?
4. In the TMS calculation, I assume that $c_{i}$ represents the embedding vector output by BERT from the decomposed claim? Is this a fine-tuned BERT model?
5. Since all the models present in Figure 3 are trained on different datasets, I wonder if by any chance any of them is trained on DSM-5. This will make a significant impact on the model accuracy.

**Strengths:**

1. Very well-drawn Figure 2, including all the details of the prompt template, making the framework easy to reproduce.
2. The proposed method is fairly novel, and is tested on several available LLM models.
3. A sensitivity analysis is performed on the parameters $\alpha$ and $\lambda$ to show how they impact the DCS, KAS and LCS.

**Summary Of The Paper:**

This paper proposed a two-stage diagnostic framework to enhance the transparency, trustworthiness and reliability of the diagnostic. The framework consists of evidence - guided diagnostic reasoning and a diagnostic confidence score module to evaluate the accuracy and logical consistency the diagnoses. The results show that EDGR outperforms the existing direct in-text prompting methods by a large margin, and shows promising clinical use for AI-assisted diagnosis.

**Weaknesses:**

1. There are some typos (Fig.2) in the paper, please proofread.
2. A lot of details are missing from the paper.
3. Some of the data reported in the result section is not consistent.

---

### Official Review · Reviewer_XZQ2 · 2025-07-14
**This paper presents a promising framework that combines evidence-guided reasoning and layered verification to improve the trustworthiness of LLM-based depression diagnosis.**

**Confidence:** 3
**Clarity Of Writing:** good
**Clinical Significance:** great
**Methodological Novelty:** good
**Overall Rating:** 6
**Final Rating:** 7

**Experiments And Results:**

great

**Questions For The Authors:**

1. Have you considered evaluating the EGDR framework on other clinical or mental health datasets to assess its generalizability beyond the D4 dataset?
2. Is there any particular reason to exclude the widely used open-source models such as LLaVA or Qwen, and how might their inclusion impact the performance or insights of your framework?
3. While the ablation study on parameters $\alpha$ and $\lambda$ (in Sec. IV-C) is informative, can you clarify how stable your framework is, when applied to varied datasets or diagnostic criteria beyond depression?

**Strengths:**

1. The paper effectively combines evidence-guided diagnostic reasoning with DSM-5 criteria, enabling interpretable and clinically grounded outputs through a dual-layered verification framework (KAS and LCS).
2. It demonstrates strong empirical performance across multiple LLMs using the D4 dataset, with consistent gains in accuracy and F1 scores, which are particularly highlighting the robustness of the EGDR pipeline in improving diagnostic reliability.

**Summary Of The Paper:**

This paper presents a structured and interpretable framework that enhances LLM-based depression diagnosis by combining evidence-guided reasoning with dual-layered validation. The empirical results are promising, particularly for domain-specific models.

**Weaknesses:**

1. The evaluation is restricted to the D4 dataset, which may limit the generalizability of the proposed framework to broader clinical contexts or other medical domains.
2. The study evaluates a few language models, but omits strong open-source contenders like LLaVA or Qwen. Including them could provide deeper insights into model generalizability.
3.  The paper primarily compares EGDR against Direct and CoT prompting; inclusion of more baseline methods would strengthen the empirical claims.
4. The related works section is relatively brief and could benefit from citing additional recent studies on interpretable LLMs and mental health diagnostics. For example:
(a) Ferrario, Andrea, Jana Sedlakova, and Manuel Trachsel. "The role of humanization and robustness of large language models in conversational artificial intelligence for individuals with depression: a critical analysis." JMIR Mental Health 11 (2024): e56569.
(b) Omar, Mahmud, and Inbar Levkovich. "Exploring the efficacy and potential of large language models for depression: A systematic review." Journal of Affective Disorders 371 (2025): 234-244.
5. There are occasional issues with writing clarity and formatting, for instance, excessive and aggressive vertical spacing (in Tables III and IV) disrupts the visual flow of the text.

---

### Official Review · Reviewer_vBAW · 2025-07-16

**Confidence:** 3
**Clarity Of Writing:** fair
**Clinical Significance:** great
**Methodological Novelty:** great
**Overall Rating:** 7

**Experiments And Results:**

great

**Questions For The Authors:**

1.	How such LLMs can assist in clinical decisions or be integrated in clinical settings can be discussed.
2.	Future work may be added to explain the next steps of the study to further improve performance.

**Strengths:**

The proposed approach integrates a unified framework for enhancing and evaluating LLM-based diagnostic reasoning.

**Summary Of The Paper:**

This study proposed a diagnostic framework using a large language model for depression diagnosis.

**Weaknesses:**

It is not easy to follow with many abbreviations.

---

### Official Review · Reviewer_bt7f · 2025-07-18
**Nice paper, introduces interpretable framework grounded in DSM-5 that significantly improves diagnostic reliability across LLMs**

**Confidence:** 4
**Clarity Of Writing:** great
**Clinical Significance:** great
**Methodological Novelty:** excellent
**Overall Rating:** 8
**Final Rating:** 8

**Experiments And Results:**

excellent

**Questions For The Authors:**

Could this framework extend to other DSM-5 categories (like anxiety, PTSD)?

How interpretable are the DCS scores for clinicians?

**Strengths:**

There is a strong methodological contribution by integrating DSM-5 criteria directly into a two-stage diagnostic pipeline, which has the potential to enhance both the interpretability and clinical grounding of LLM-based mental health diagnosis. The proposed Evidence-Guided Diagnostic Reasoning (EGDR) framework and the Diagnosis Confidence Score (DCS) introduce symbolic reasoning into what is typically a black-box process, which is novel considering the clinical NLP space. The experimental results show significant improvements in both diagnostic accuracy and reasoning quality across multiple LLMs. These findings demonstrate clear potential for real-world deployment, especially in supporting mental health screening in resource-constrained settings

**Summary Of The Paper:**

The paper proposes a two-stage LLM-based diagnostic system for depression: (1) Evidence-Guided Diagnostic Reasoning (EGDR), which guides reasoning using DSM-5 criteria, and (2) a Diagnosis Confidence Scoring (DCS) system to evaluate factual alignment and logical consistency. The framework is tested on the D4 dataset.

**Weaknesses:**

The paper relies a bit heavily on pseudo-labels generated by GPT-4o-mini, which may limit the objectivity or generalizability of the results, especially in the absence of expert clinical validation. The proposed system also assumes access to a structured DSM-5 knowledge graph, which might not be easily replicable in practice or across other diagnostic domains.

The paper could benefit from additional discussion on how interpretable or actionable these scores are for practicing clinicians.

---

### Official Review · Reviewer_aeWD · 2025-07-18
**Leveraging Evidence-Guided LLMs to Enhance Trustworthy Depression Diagnosis**

**Confidence:** 4
**Clarity Of Writing:** good
**Clinical Significance:** good
**Methodological Novelty:** great
**Overall Rating:** 5

**Experiments And Results:**

great

**Questions For The Authors:**

How does performance compare when evaluating against expert‑labeled ground truth rather than pseudo‑labels?
Confirming similar gains with human annotations would validate clinical utility; significant discrepancies might require rethinking reliance on model‑generated labels.

Have clinicians reviewed a sample of EGDR outputs and DCS scores to assess hallucination rates and real‑world plausibility?
Low hallucination and high clinician agreement would boost confidence in adoption; evidence of medically incorrect reasoning would necessitate tighter grounding or hybrid methods.

Does the EGDR + DCS framework generalize to other psychiatric or medical diagnoses (e.g., anxiety, bipolar disorder)?
Demonstrated cross‑domain applicability would elevate the framework to a broadly useful tool; limited success might confine it to depression only.

What is the end‑to‑end computational cost (API calls, GPU hours) for generating EGDR outputs at scale, and have you evaluated lighter‑weight LLM alternatives?
Feasible runtime and cost would support clinical deployment; prohibitive expense could drive exploration of distilled or open‑source models.

Have you analyzed DCS stratified by patient demographics to check for potential bias amplification?
Uniform performance across groups would ensure equitable care; disparities might prompt integration of fairness constraints or demographic covariates.

**Strengths:**

The structured two‑stage design combines the generative power of modern LLMs with a formal DSM‑5 knowledge graph, transforming opaque “black‑box” outputs into semi‑structured hypotheses and interpretable scores. Empirical gains, up to +45% accuracy and +36% DCS, are striking and consistent across multiple models, underscoring the generality of EGDR. The introduction of KAS and LCS offers clear, actionable insights into why a diagnosis is or isn’t trustworthy, which is essential for clinician acceptance. Thorough ablations on α and λ parameters strengthen the methodological rigor, and the case studies concretely demonstrate the framework’s diagnostic fidelity.

**Summary Of The Paper:**

The authors present a two‑stage framework for depression diagnosis using large language models. First, the Evidence‑Guided Diagnostic Reasoning (EGDR) procedure steers the LLM to interleave evidence extraction from patient–clinician dialogues with logical inference grounded in a DSM‑5 knowledge graph, producing semi‑structured diagnostic hypotheses. Second, the Diagnosis Confidence Scoring (DCS) module assigns each generated diagnosis a composite score, with a Knowledge Attribution Score (KAS) verifying claim‑level alignment to DSM‑5 triplets and a Logic Consistency Score (LCS) checking adherence to formal diagnostic rules, normalized into a single interpretable metric
.

Evaluated on the D4 depression dialogue dataset using GPT‑4o‑mini pseudo‑labels, EGDR outperforms both direct prompting and chain‑of‑thought across five LLMs, achieving up to a 45‑point accuracy boost and 36‑point DCS improvement on OpenBioLLM. Ablations over the α (semantic vs. symbolic attribution) and λ (KAS vs. LCS weighting) parameters demonstrate the stability and interpretability of DCS. Two detailed case studies illustrate high‑ and low‑confidence scenarios.

**Weaknesses:**

The entire evaluation hinges on pseudo‑labels generated by a model rather than human expert annotations, leaving open the question of real‑world correctness and potential teacher‑model bias. No clinician review of the generated reasoning or confidence scores is reported, so hallucination rates and clinical plausibility remain unquantified. The study is confined to a single dataset (D4) and a single condition (depression), limiting evidence of generalizability to other mental health disorders or medical domains. Computational cost and latency of GPT‑4‑based prompting at scale are not discussed, nor is any fairness analysis across demographic subgroups to guard against bias amplification.